# Hydrostratigraphy and hydrogeophysical studies to delineate fresh and saline aquifer boundaries in Lesser Cholistan of Pakistan

**Kashif Arif, Perveiz Khalid** *

Institute of Geology, University of the Punjab, Lahore, Pakistan

* perveiz.geo@pu.edu.pk

## Abstract

The differentiation of saline water and fresh water interfaces is a key objective in ground water exploration and management. Bahawalpur is the twelfth biggest metropolitan area of Pakistan situated in south Punjab near the bank of River Sutlej and lies at 29˚59'55" N latitude and 73˚15'12" E longitude at an elevation of 521 ft AMSL in the Cholistan area close to the Thar abandon. The study area comprised of Lesser Cholistan experiencing acute shortage of water for inhabitants and livestock as well. The occurrence of fresh water is also challenging because of high salinity in groundwater. The present study is intended to identify hotspots of fresh groundwater zones. To achieve the goal, vertical electrical resistivity and borehole data are used to mark fresh and saline interfaces in groundwater. To achieve the results 230 vertical electric sounding were performed in the study area. A total of 3 to 5 geo-electric layers are identified with modeling along with the processing and interpretation of resistivity data. In the study area, resistivity values are classified as very high (>230 Ω-m), high (230–100 Ω-m), medium (100–40 Ω-m), low (40–20 Ω-m) and very low (<20 Ω-m). Borehole data is used to interpret subsurface lithologies and to calibrate the modeled resistivity curves. The electric resistivity data indicates that thick layers of Quaternary sediments is present in the subsurface that is primarily composed of clay, silt, sand, gravels and some kanker. Inversion technique is applied to generate 2D subsurface resistivity maps to delineate fresh and saline water zones. The generated 2D resistivity maps at variable depth above and below water table and formation resistivity maps are successfully utilized to differentiate fresh and saline water zones. The identification of a saline water aquifer within sediments rich in clay was made possible by the observation of very low resistivity measurements in the southern region. Conversely, the detection of relatively high resistivity values, coupled with the presence of sand and gravel deposits in the northern section of the lesser Cholistan area, provided compelling evidence of the existence of fresh groundwater. These findings have significant implications for the management of water resources in the region, as they provide valuable insights into the distribution and availability of groundwater resources for future use.

**Data Availability Statement:** "All relevant data are within the paper and its Supporting Information files."

**Funding:** The author(s) received no specific funding for this work.

**Competing interests:** The authors have declared that no competing interests exist.

## Introduction

The sustainable supply of freshwater is one of the key parameters in urbanization and the socio-economic development of an area [1, 2]. Surface water, which includes streams, canals, lakes, rivers, etc. and groundwater are two primary cheap and easily available sources of freshwater [3]. Due to high solubility of different solutes water is capable to dissolve suspend, soak and adsorb various impurities such as arsenic, sulfides, and chlorides, organic and inorganic etc. [4]. Pakistan is among those countries that are facing freshwater scarcity. The whole country is situated in arid to semi-arid regions with low precipitation. The majority of the population relies on groundwater for their domestic, agricultural and industrial needs. The over-extraction of groundwater is one of the main causes of increasing salinity in the groundwater whereas the poor dumping strategies of solid waste is the main source of groundwater contamination [5].

Cholistan desert–a part of Thar Desert–covers an area of 25,800 km$^2$ and is the largest desert in the southern Punjab province of Pakistan (Fig 1). This desert is divided into two parts: the Lesser Cholistan and the Higher Cholistan. Around 16,000 km 2 of the 47 district is comprised of the Cholistan Desert that is covered by dunes of low height. The density of dunes increases towards the northwest [6]. Bahawalpur City is the largest city in the Lesser Cholistan while the Higher Cholistan is uninhabited with high sand dunes up to 100 m. The economy of this district mainly based on agriculture thus the continuous supply of freshwater is the backbone of its economy. However, the existing groundwater is contaminated by hazardous pollutants such as exceeding amount of different cations, anions, TDS, total hardness and arsenic. The analysis of groundwater performed by PCRWR from 2002 to 2015 demonstrates that the groundwater in most of the localities in this district is not safe for drinking. In the southern part of the Bahawalpur area, between Fort Abbas and Derawar Fort, there is a zone of reduced salinity. This relatively low-lying area corresponds to the ancient channel of the Ghaggar (Hakra) River. Although the Ghagger (Hakra) channel is now generally dry, floodwaters occasionally flow through it from the east, near Fort Abbas, in periods of uncommonly high rainfall. Available data indicate that the upper part of the aquifer along this channel contains water of about 4,000 ppm TDS. A deep-water sample (~ 410 m depth) collected near to Derawar Fort contained high salt concentrations up to 25,400 ppm [7].

The study area is a part of the Punjab Platform that is a subdivision of the Lower Indus Basin. The study area consists of alluvial and aeolian plains [6] of Lesser Cholistan. The area along the Sutlej River comprises the alluvial plain whereas dunes cover the aeolian plain of the Desert. The planning and management for better quality water supplies are in efficient in Bahawalpur City [8]. The use of hydrogeological investigations aided by appropriate geophysical techniques to assess the groundwater potential in the confined or unconfined aquifer systems in a particular area is very common [9–22]. The spatial and vertical distribution of different characteristics of the aquifer such as facies distributions and the variation hydraulic parameters variations are fundamental for the study of alluvial deposits.

Since the saline water has an enormous amount of total dissolved solids (TDS), modeling and delineation of fresh and saline water boundaries in the subsurface are necessary before planning depths and locations of water extraction wells and tube wells. The Sutlej River is at western periphery of the study area (Fig 1) which is believed to control the subsurface paleo-channels. The primary objective of this study is to delineate fresh and saline water interfaces and to map the fresh water aquifer boundaries. With the help of results of the present study, the exploitation of fresh groundwater with optimized location of water-wells installation would be possible which will further lead to the development of such a desertification of the study area. As the reported salinity values of groundwater of Lesser Cholistan are on higher

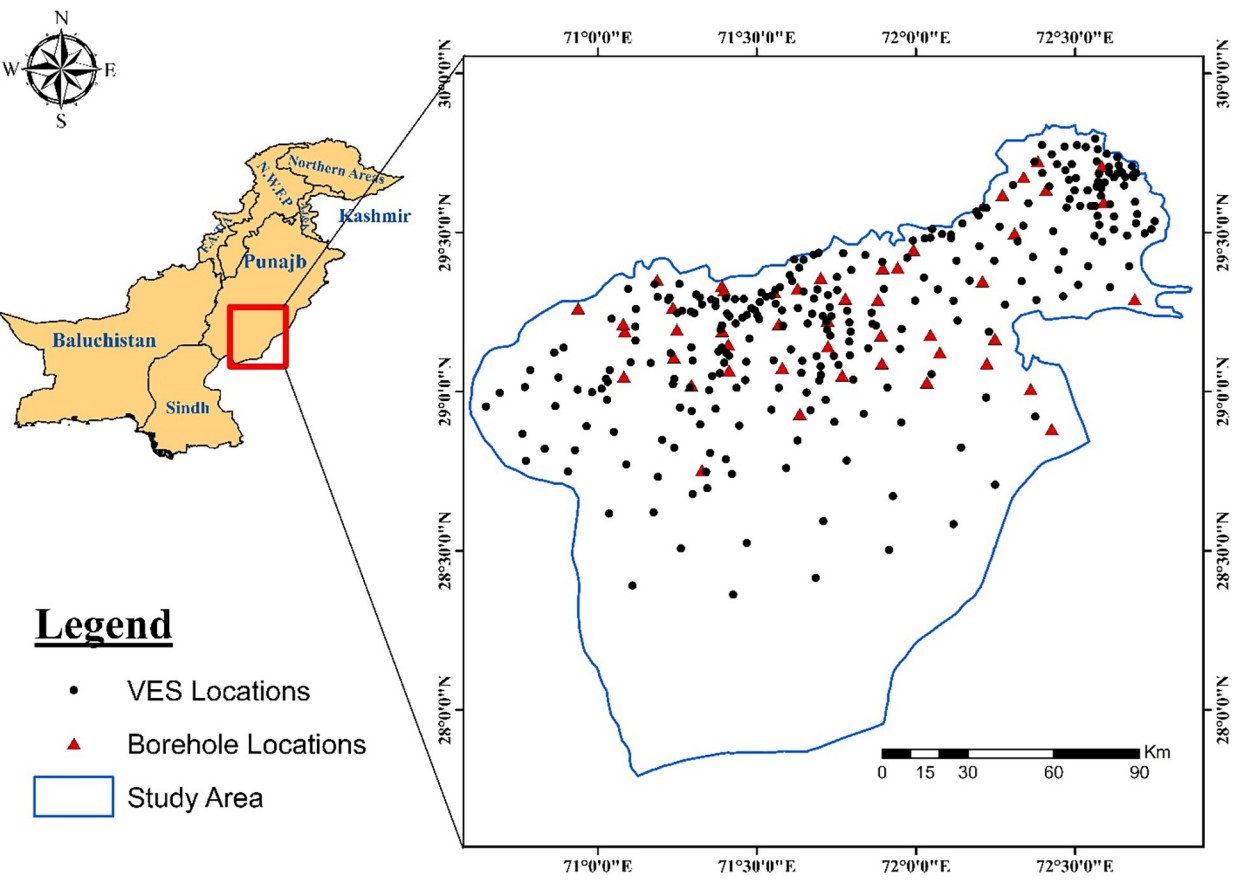

**Fig 1. Base map of the study area showing VES and Borehole locations.**

side. So, this study has its own importance to exploit fresh water pockets within the saline-brackish groundwater zones.

## Geology and geomorphology of the area

Pakistan is situated at the collisional boundary of the Indian and Eurasian Plates, which creates a significant number of intermontane basins and valley systems [20]. The Indus basin is one of several basins produced by the action of tectonic activity. Sediments generated by the erosion of the surrounding mountains are transported into this basin via several sedimentary transport agents. These sediments may vary in size from very coarse alluvial fan gravels and channel deposits with high hydraulic conductivities to very fine lacustrine clays with low hydraulic conductivity. Thus, the grain size of the sediments, lithology and geomorphic nature of the deposited sediments are the important factors in the development of clastic fluvial aquifers and their hydraulic characteristics [23, 24]. Since the study area is far away from the source of the sediments mainly fine-grained silt/clay and medium to coarse-grained sand are present predominantly. The fine sediments of clays/silts are believed to be exist due to the episodically weathered subsurface sedimentary succession especially Siwaliks [6].

The borehole data suggests that a 457 m thick cover of alluvial sediments comprising of fine-grained silt and clay to medium to coarse sand of Late Quaternary age is present throughout the area [25]. This cover is heterogeneous in nature with significant lateral and horizontal variations in lithology [26, 27]. Thus, multiple aquifer systems may be present in the study

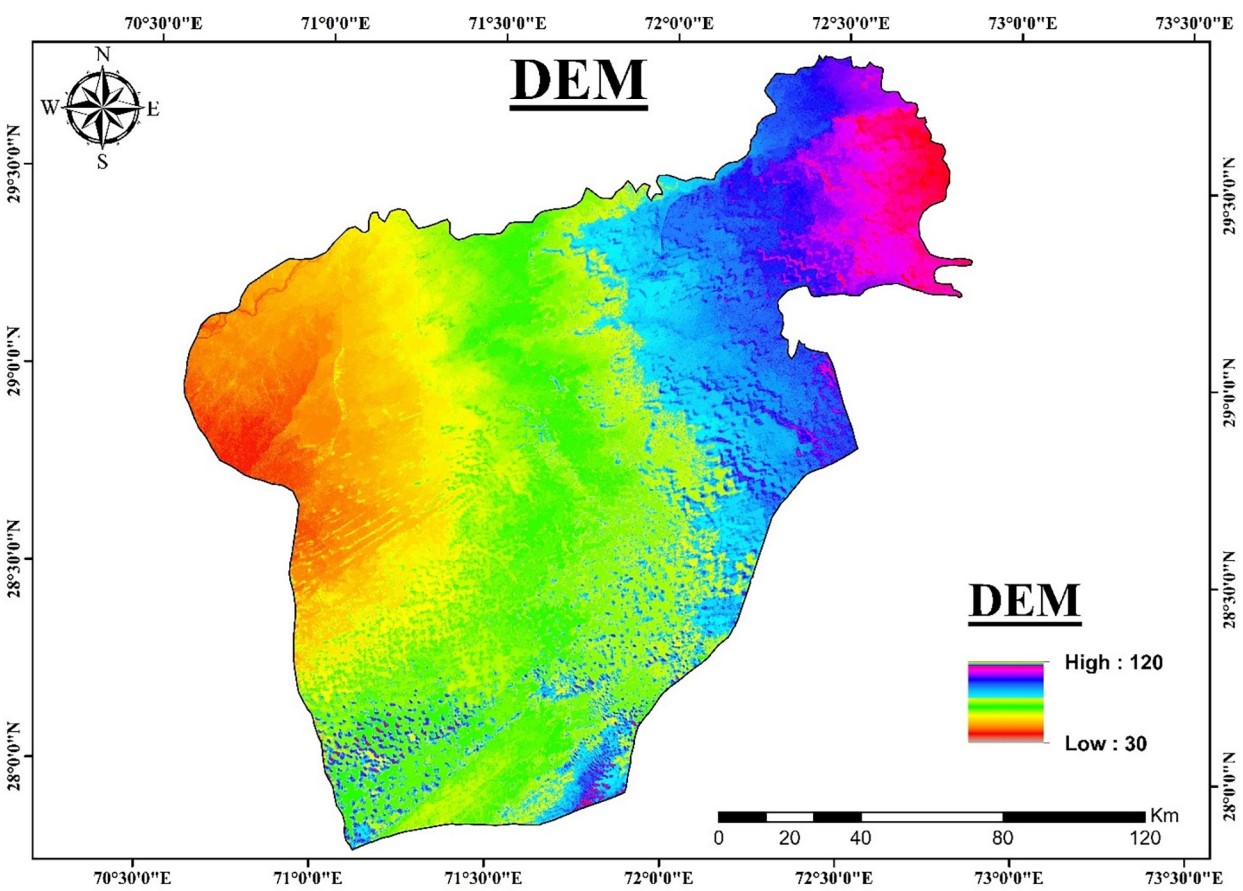

**Fig 2. Digital elevation model shows major geomorphology of the area.**

area. According to [28] report, the alluvium is saturated up to 1,000 m depth. No continuous layer of clay is reported above the aquifer, thus an unconfined type of aquifer may be present in the area. However, confined or semi-confined aquifer conditions may prevail locally. The landforms are developed primarily by the action of river deposits and later on by wind deposits of sand dunes. The landform of the study area consists of two units: flood plain deposits and aeolian deposits. The width of flood plains varies from 10–20 km in different localities. According to geomorphological classification, the study area consists of level plains, channel infills, basin, and levees as shown in Fig 2.

The sediments deposited in the area are mainly fine to medium grained silt, clay, and sand in the association of kanker. Kankers are basically stiff pebbles usually made up of mud or silt. The sand plains of the Cholistan Desert are comprised of medium-height sandy ridges and inter-dunal hollows of the Hakra River. These deposits are composed of fine to medium silty sand and silt with low permeability. These grains are well rounded, well-sorted, and transported by wind from adjacent arid zones (Figs 3 & 4). An old channel of the Hakra River forms a freshwater channel [7]. In the 16th century, this river ceased flowing [29] and until the mid-20th century, the river flowed only during heavy rainfall events. This ephemeral fluvial environment exhibits a flat, hardpan surface covered by irregular patches of dunes up to 10 m high. The upper layer of the floodplain comprises fine-grained silt to sandy loam and silty clays of low permeability but good porosity. Thus, rainwater may accumulate in depressions and may stay for a couple of days to several weeks. Since 1960, some portion of this desert has

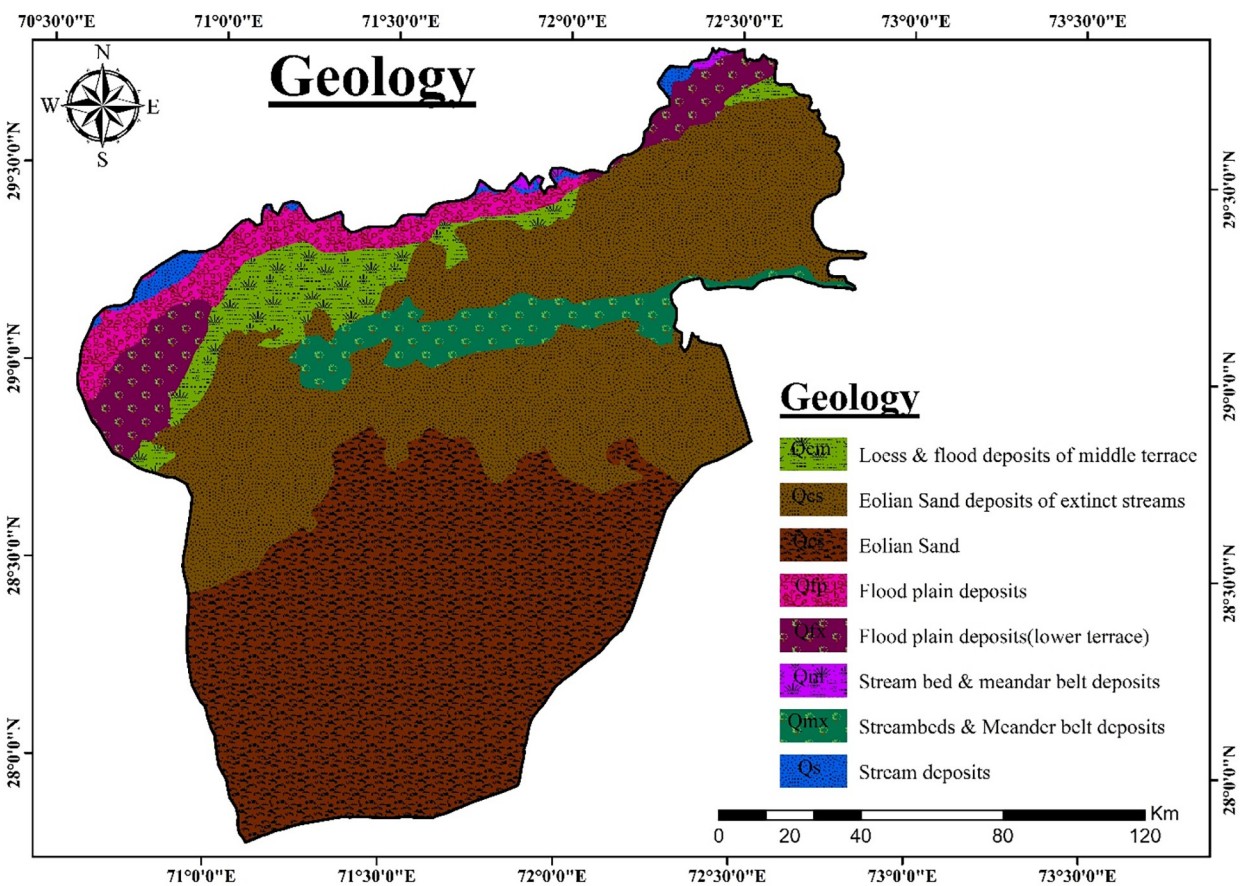

**Fig 3. Geological map of Cholistan desert.**

been converted into cultivated land after the construction of irrigation canals from Sutlej River.

Rainwater and the Sutlej River are the main recharge sources in the study area. The area receives less than 200 mm of rainfall annually. Due to the lack of permeability the surface, rainwater does not penetrate deeply into the underlying sediments. The recharge rate can be divided into deep and shallow categories [7]. The palaeohydrological situation can be used to explain different recharge rates for shallow and deep aquifer layers. No reliable temporal information is available in the western part of the Cholistan area. However, pluvial conditions existed between 13 000 and about 4000 years ago in the eastern part of the Cholistan [30–32]. According to literature, two pluvial periods may have existed from 13,000 to 8,000 years and 7,000 to 4,000 years [7].

A few studies related to hydrogeophysical investigations and groundwater quality are available in the literature. In 1968, the USGS conducted a study on groundwater hydrology of the Punjab Province. According to this study, Most of the area of district Bahawalpur comprises highly saline water with TDS> 5000 ppm except a narrow freshwater corridor along the Sutlej River and the old channel of Hakra River between Fort Abbas to Derawar Fort. Geyh and Ploethner [7] conducted a study on physio-chemical analysis of shallow water samples collected from 20–70 m depth. They concluded that the groundwater is highly mineralized in most of the area of Rahim Yaar Khan District with a high concentration of TDS (27,000 ppm) and high sodium absorption ration ~ 54 mg/l).

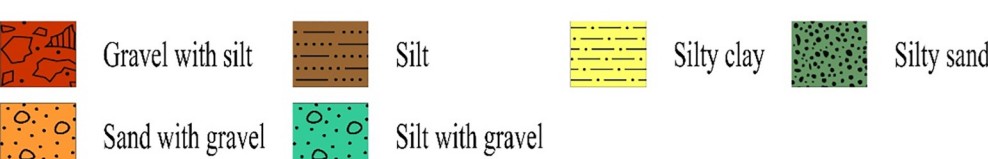

**Fig 4. Boreholes in the study area representing the shallow subsurface geology.**

## Methodology

Electrical resistivity has long been used as a method to detect subsurface hydrological and geologic features with considerable success [9, 10, 14, 21, 33]. Electrical resistivity depends on both the lithologic makeup of the aquifer and upon the fluid contents of the aquifer [22, 34, 35]. The methodology used in this study involves assessment of existing data and the acquired data set. The acquired data comprised of vertical electrical soundings (VES) up to 200 m depth at sparsely distributed locations. For convenience, the VES points were generally collected at distance of 4 to 8 km, mostly along the roads, canals and existing tracks as shown in Fig 1. A terrameter SAS 4000 is used for VES data acquisition. All VES data were acquired by vertical sounding using a Schlumberger array configuration. The Schlumberger array consists of four electric conductors known as electrodes. The two end conductors are current electrodes and the inside two conductors are the potential electrodes. The potential electrodes are installed at the centre of the electrode array with a small separation, typically less than one-fifth of the spacing between the current electrodes (AB $\geq$ 5MN). The current electrodes are increased to a greater separation during the survey while the potential electrodes remain in the same position until the observed voltage becomes too small to measure.

The data from approximately 120 test holes and 20 test wells up to 300 m depth is available from the Water and Power Development Authority (WAPDA) of Pakistan. More than 230 vertical electrical resistivity sounding (VES) data are acquired in different places of the study area was used for the characterization of various subsurface lithologies. Initially the field VES data were processed using curve matching technique in which the field graphs/curves were compared with sets of standard curves. Among the various curve matching techniques, partial curve matching technique with auxiliary point method was utilized to process the field data. Ebert auxiliary standard graphs were used for this purpose. However, the final interpretation was made by using a combination of curve matching technique and computer iterative modeling to find out number of geo-electric layers in the study area. The analysis of VES curves is difficult task and requires that the measured curve be matched with several model curves and each model curve represent different subsurface resistivity distributions. To select the final model is constrained with the help of VES data, lithologs, local geological data, and borehole data of study area, we map different depositional zones in the Bahawalpur district to mark the alluvial aquifer systems as well as the salinity of the groundwater. For calibration and verification of VES results with borehole data, some of the VES points are acquired near to the existing boreholes/tubewells. Furthermore, a lithological borehole was observed by placing it alongside a processed VES field curve representing the all subsurface geo-electric layers with their true resistivities/depths and locating closest to that specific borehole. All the VES data points were modeled (Fig 5) by using computer software IP2Win [36–39], using information derived from lithology logs, ground water levels, geologic maps and EC maps. 2 D models are generated and plotted against the resistivity values verses depth and consist of several horizontal layers separated according to discrete band of resistivity as shown in Fig 5A–5F. In general calibration between lithology and resistivity is established using available borehole information and resistivity data and is shown in Table 1 and in Fig 6A. Fig 6B exhibits the geographic location of calibrated VES curves and borehole logs.

The resistivity classification provided in Table 1 was established after analyzing integrally the results of all VES and litho-logs in the study area keeping in view the geology of the study area using the reported literature. Each VES model curve represents the virtual VES response of horizontally stratified earth layer using a limited number of sedimentary layers (3–5). The base layer in each model extends to undefined depth value.

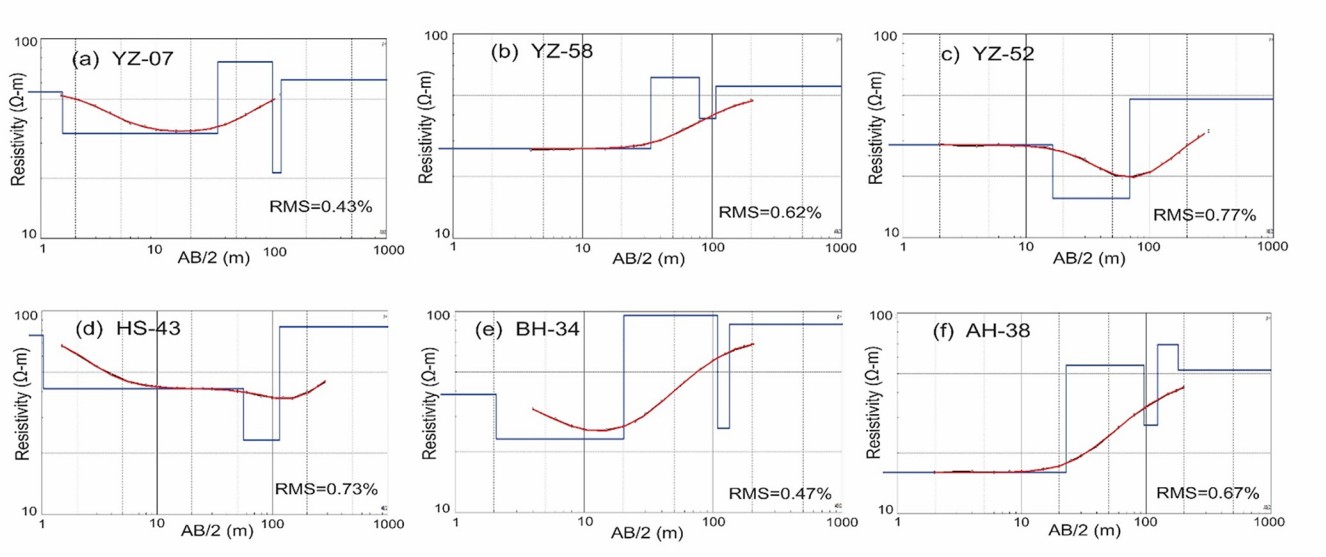

**Fig 5.** a-f: Apparent resistivity data are marked by small circles. Solid black curve represent the apparent resistivity curve. Red curve is the best fitted curve to the apparent resistivity data. Solid blue block line is the modeled resistivity (synthetic resistivity). Horizontal axis is the current electrode spacing (AB/2) in meters and vertical axis is the resistivity in ohm meters.

## Result and discussion

Interpretations are made for three to five geo-electric subsurface layers in the study area. The resistivity and depth/thickness of each layer is distinctive. Electrical resistivity values of each geo-electric layer can be implemented to characterize the probable underlying lithology, including particle size, porosity, and fluid type. A correlation between the interpreted underlying lithology and true resistivities is created by simultaneously examining the processed data from 230 VES probes and lithological logs of boreholes (Table 1). Furthermore, the lithological logs of boreholes closest to the respective VES probes are used to calibrate the true resistivities (Fig 6A).

Clay, silt, and sand comprise the alluvial deposits with gravel and pebbles are encountered in boreholes in the aquifers. The southern portion of the lesser Cholistan has clay-rich sediments as a result of the intrusion of salt contents, and the findings of the resistivity inversion provide confirmation of this. The interpretation of VES results in this area was challenging due to the relatively low difference in resistivity readings between shallow and deep alluvium. Sand and gravel sediments serve as a good aquifer material for fresh groundwater and clay rich sediments serves as a saline water aquifer, according to borehole data shown in Fig 4. The digital elevation map of Cholistan, indicates that the elevation declines from northeast to

**Table 1. Cutoff values of resistivity for different lithologies used in interpretation of VES data.**

| Resistivity Zone | Resistivity Range (Ω-m) | Interpreted Lithology |
|---|---|---|
| Very high | Greater than 230 | Coarse sand with gravels and kanker |
| High | 230 to 100 | Sand with gravels with minor interbeded thin layer of silt and clay |
| Medium | 100 to 40 | Medium to coarse sand |
| Low | 40 to 20 | Mixture of sand with silt and clay |
| Very low | Less than 20 | Silty clay / clayey silt (Above GWT)<br>Saline sediments, silty clay in dominance (Below GWT) |

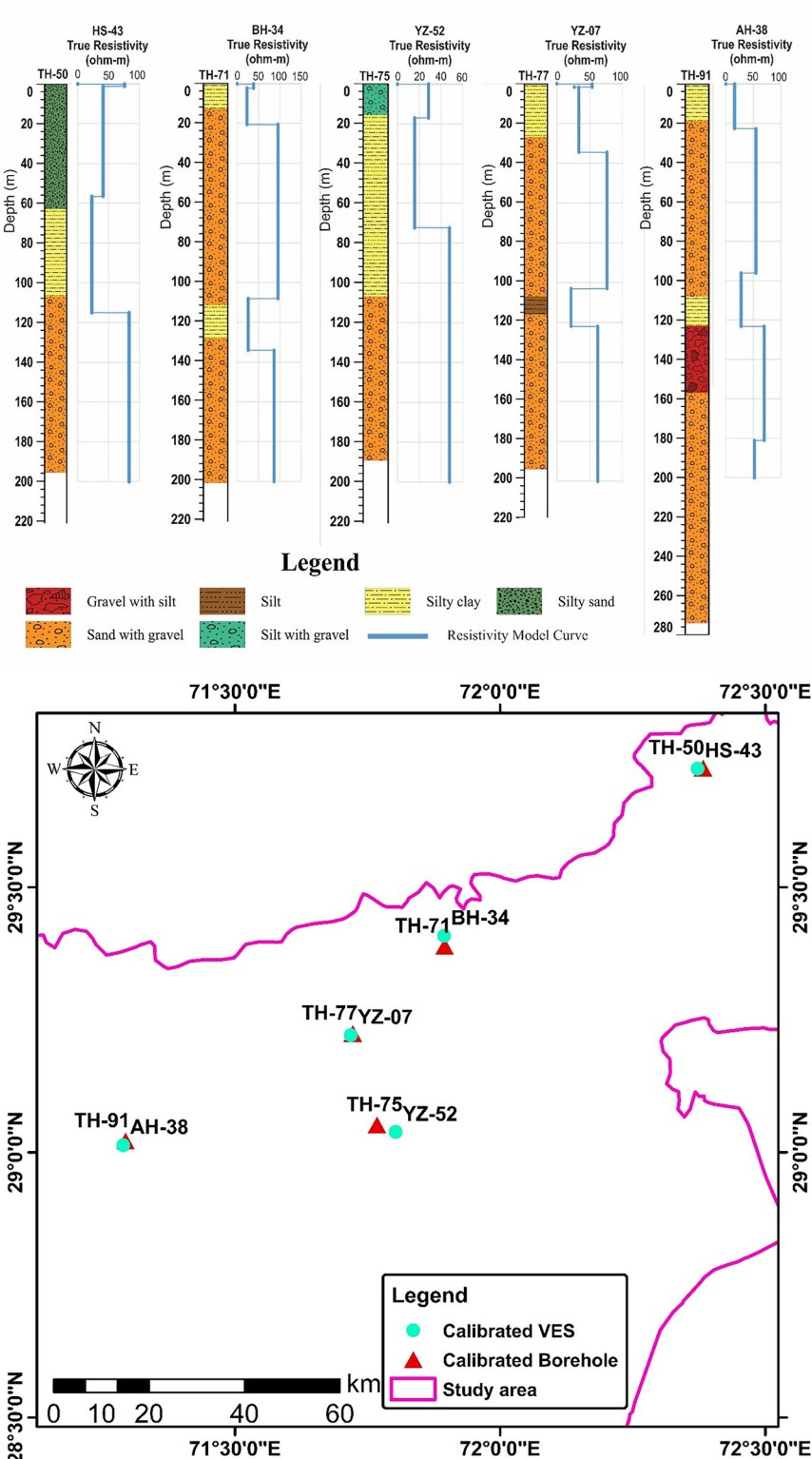

**Fig 6.** a. Borehole logs and modeled resistivity curves representing the calibration between lithology and resistivity data. b: Map depicting spatial location of calibrated Borehole logs and modeled resistivity curves.

southwest, the streams run in a north to southwest direction (Fig 2). The digital elevation map and the water table elevations are interrelated. The ground water level changes across the Lesser Cholistan. In the northeast at well TH-50, the water table is around ≤ 12m deep, while in the southwest borehole TH-91, it is about ≤ 17m deep. As can be seen in Fig 7A–7H, the electrical resistivity survey findings indicate that the south portion of the Lesser Cholistan has significantly lower resistivity values than the north side. In the south region of the study area (Fig 7), where the depth of current penetration in the VES findings is very low, resistivity data results confirm that higher proportions of clay-rich or salty sediments prevent the electric current from flowing at greater depths. The borehole lithological logs also support the argument. The dry sand, gravel, and boulder sediments are linked with high resistivity values that are above the water table and close to the surface, whereas the freshwater saturated with sand-gravel sediments are linked with high resistivity values that are below the water table. The primary aquifer in the region, which is largely on the north and northwestern side of the area, is mainly composed of water-saturated sand-gravel sediments with resistivity values between 20 and 200 $\Omega$.m. The saline sediments are predominantly found on the research area's southern side, with resistivity values < 10 $\Omega$.m especially below the water table. These low values of resistivities are interpreted as clay/silt-rich sediments which restrict the groundwater flow due to low permeability and eventually leads to the salinity of groundwater in the region. The research region is segmented into saline and fresh water aquifers because of the salinity factor.

To study the fluctuation in resistivity values in the lesser Cholistan, a variety of resistivity maps are prepared at various depths, above and below the water table, and these maps are displayed in Fig 7A–7H at depths of 5-, 25-, 50-, 80-, 100-, 125-, 150-, and 200-m, respectively. These depth levels are chosen after a comprehensive evaluation of the resistivity data to identify the areas with the maximum fluctuation in resistivity. Fig 7A shows the resistivity map at a depth of 5-m, which is above the water table. Therefore, the resistivity values reflect mainly lithologic differences. The resistivity value in the research area ranges from less than 10 $\Omega$.m to more than 200 $\Omega$.m, as shown in Fig 7A. Clayey silt or silty clay sediments are indicated by low resistivity values (<10 $\Omega$.m) at a few locations along the northern, and southern sides. High resistivity readings above the water table near the surface reveal boulders and coarse sand-gravel sediments. Fig 7B displays the resistivity map of the confined aquifer, which ranges from 0 to 200 $\Omega$.m. The northern side of the research region exhibits resistivity values of 20–200 $\Omega$.m (Fig 7B). Resistivity values up to 40 $\Omega$.m suggest silty clay or clayey silt with brackish-slightly good quality ground water, while values < 10 $\Omega$.m below the water table show saline sediments. Sand-gravel with good groundwater is indicated by resistivity values greater than 40 $\Omega$.m. The lesser Cholistan's southern region has values < 20 $\Omega$.m above the water table, which represent clayey silt or silty clay. The resistivity maps for 50, 80, 100, 125, and 150 m depth levels are displayed in Fig 7C–7G. In these maps near VES locations AH-38, YZ-52, and YZ-58, the clay-silty clay and saline sediments at variable depths are interpreted which indicate very low resistivity value below 20 $\Omega$.m. The incredibly low resistivity readings in magenta color are taken to indicate the presence of a saline aquifer. These maps represent sites around the VES probes KH-22, AH-40, and HS-22–25, which are believed to represent an aquifer of fresh water since they are mostly composed of sand, gravel, and boulder sediments and have relatively high resistivity values between 60 and 200 $\Omega$.m. Maps are interpreted with resistivity values between 20–40 $\Omega$.m and below the water table at different depths to indicate areas with clayey or silty sand and marginally fit quality groundwater. Fig 7H depicts the resistivity map at a 200-m depth. The lesser Cholistan's northeastern and northwestern regions contain sand and gravel with medium to high resistivity values of 60 to 200 $\Omega$.m. These resistivity readings of 60 to 100 $\Omega$.m have been found in several southern locations close to the YZ-68,69,70,71, and YZ-73, which may yield groundwater of good quality. Resistivity values up to 40 $\Omega$.m

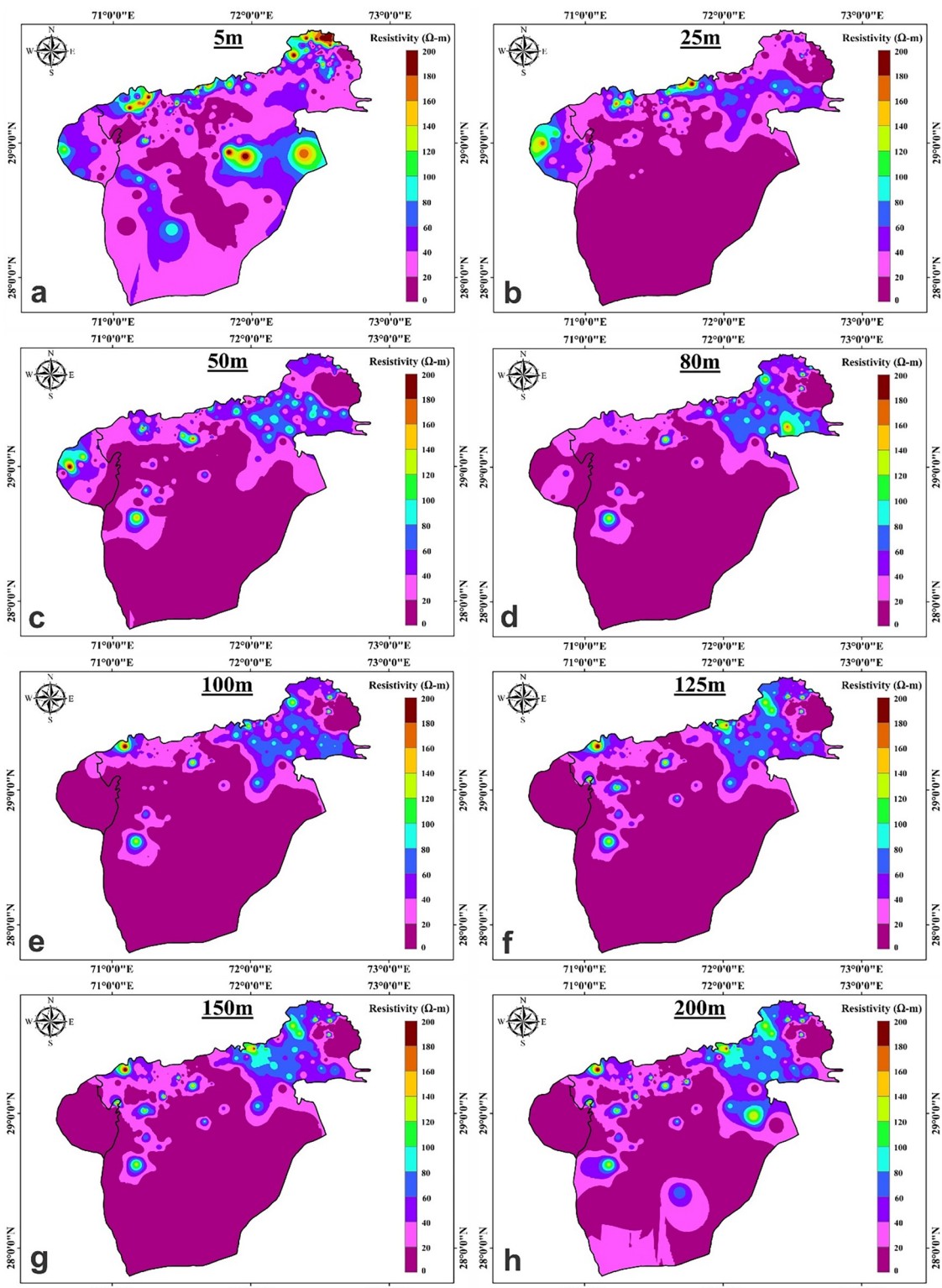

**Fig 7.** a-h: Maps depicting the distribution of true resistivity across varying depths in two dimensions.

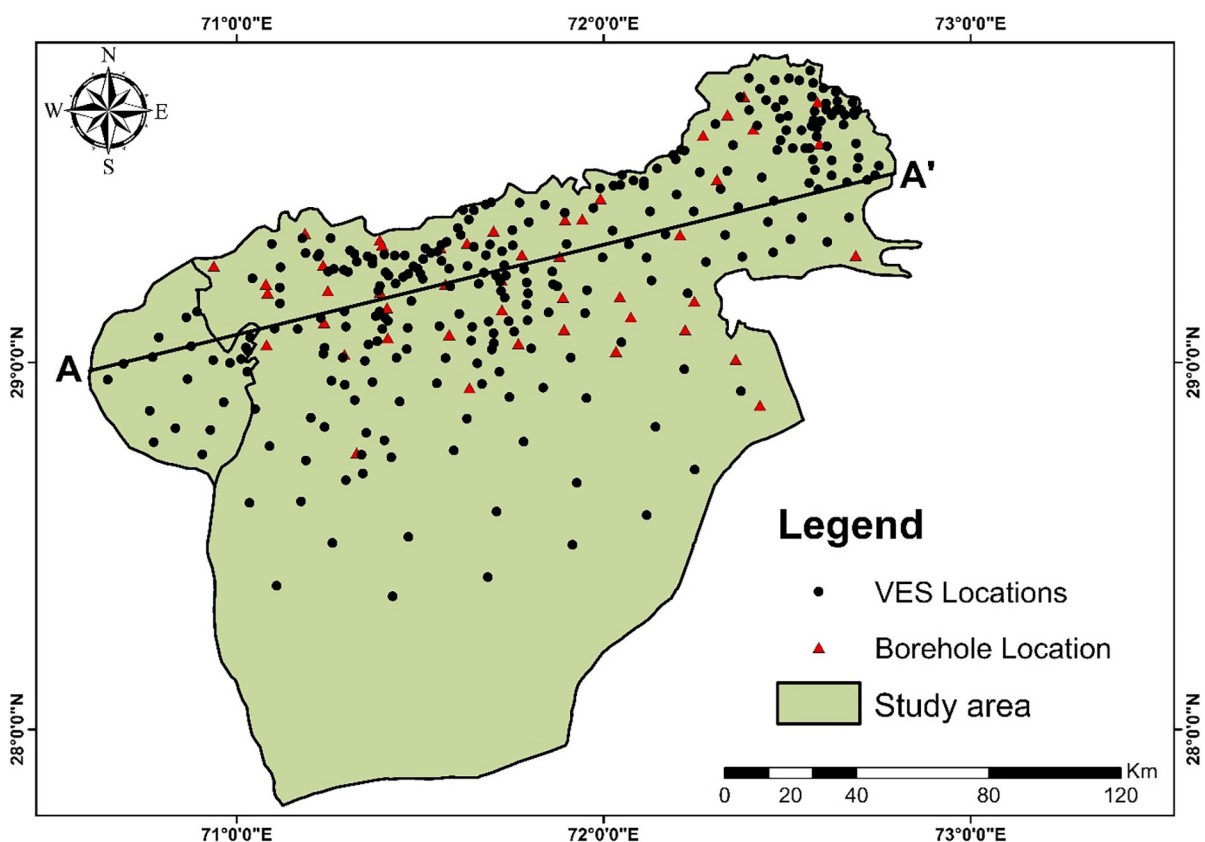

**Fig 8. Base map of the study area showing VES points, boreholes, and cross section AA'.**

suggest saline sediments, clayey silt, silt and silty sand, with brackish and marginally fit ground water. We can see from the resistivity analysis of this research area that the salty water aquifer on the southern side of the region is mostly attributable to the paleochannel dissolved with salt content. The salt content was dissolved and infiltrated to the aquifer during alluvial and fluvial deposition of sediments derived from volcanic rocks and ash exposed in the surrounding area.

In order to comprehend the hydrostratigraphy of the research region, the profile A-A′ is chosen on the base map (Fig 8). Interpolation of resistivity data along the profile AA' yields the resistivity cross-section AA' (Fig 9A & 9B). The electrical resistivity processed data and borehole data closest to the profile A-A′ are used to create 2D cross sections (Fig 9A) along this profile. The profile's interpreted outcomes are displayed in Fig 10A as a generalized five-layer scenario. Low and high resistivity values appear to fluctuate along the profile AA'. The fresh water is at a very shallow depth, as shown by the high resistivity values between VES sites KH-22 and HS-25. AA's profile's cross-section clearly reveals five layers. Above the water table, which varies roughly from the west to northeast side of the profile, is the first layer, which is composed of dry sediments like silty sand and sand-gravel. The second layer exhibits a fresh-water aquifer with some high resistivity values on the northeastern side, whereas the western side shows a sand aquifer with brackish quality water between LQ-10 and KH-22. On the western side, between LQ-5 and KH-33, there is a third layer of silty sand, and on the northeastern side, there is a sand aquifer with quality ground water. A fourth layer of clay, silty clay with considerable fine silt, can be observed from the western side, between LQ-10 and KH-22, showing resistivity values of 30 Ω.m. From the western side, a fifth layer of saline sediments

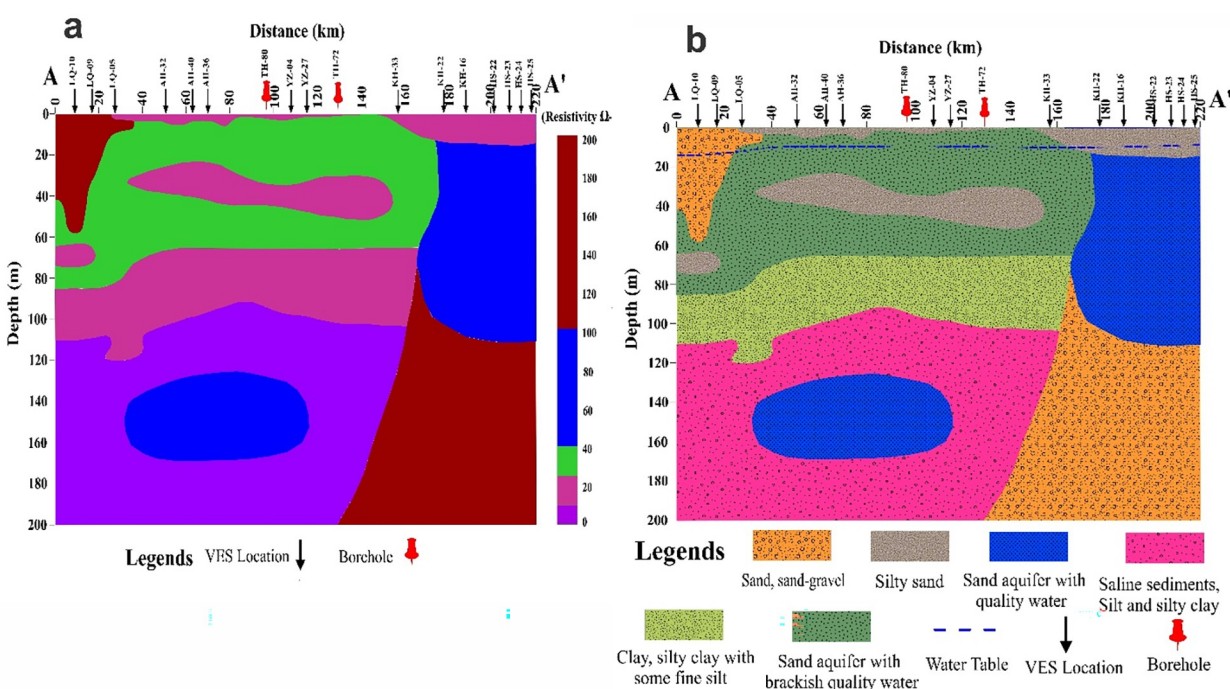

**Fig 9.** a) Electrical resistivity distribution versus depth along cross section AA', b) Lithological model of the subsurface along cross section AA'.

of ≤ 10 Ω.m can be depicted sustaining a sand aquifer between LQ-05 and YZ-27 at a depth of 135–170m. While there is sand and gravel with a high resistivity of 100 to 200 Ω.m on the northeastern side between KH-33—HS-25, this may yield good-quality groundwater. A typical stacked representation of 2D depth maps is generated to clearly depict the distribution of lithologies and aquifer conditions with depths up to 200 Ω.m (Fig 10).

## Conclusions

Groundwater salinity can be determined using a VES survey, which can also be used to detect subsurface aquifer layers. In the study area, the VES survey was carried out in 230 pre-selected locations. The groundwater aquifer conditions in the lesser Cholistan area of the district of Bahawalpur were evaluated spatially using the VES findings and lithological logs of boreholes. In the research area, three to five geo-electric subsurface layers with varying thicknesses are evaluated as an outcome of processing geophysical data. The resistivity values are classified as very low (<20 Ω-m), low (40–20 Ω-m), medium (100–40 Ω-m), high (230–10 Ω-m) and very high (>230 Ω-m). A number of resistivity maps are generated at different depths, both above and below the water table. The distribution of fresh and saline water aquifers in the study area is determined by comparing all of the electrical resistivity maps. The modeling and re-inversion of 1D electrical resistivity data collected in the lesser Cholistan delineated the boundary between the saline water and freshwater zones, with remarkable achievement. Borehole data from several localities throughout the region were used to map the near surface lithology and to calibrate the modeled resistivity curves. At each resistivity point, the value of the formation resistivity was calculated and validated using the lithologies of drilled boreholes. The 2D resistivity surface maps enabled in describing the fresh, marginally fit and saline water zones at various depth levels. Most of the study area with very low resistivity values especially the southern portion indicates the occurrence of a saline water aquifer of clay rich sediments. On the other

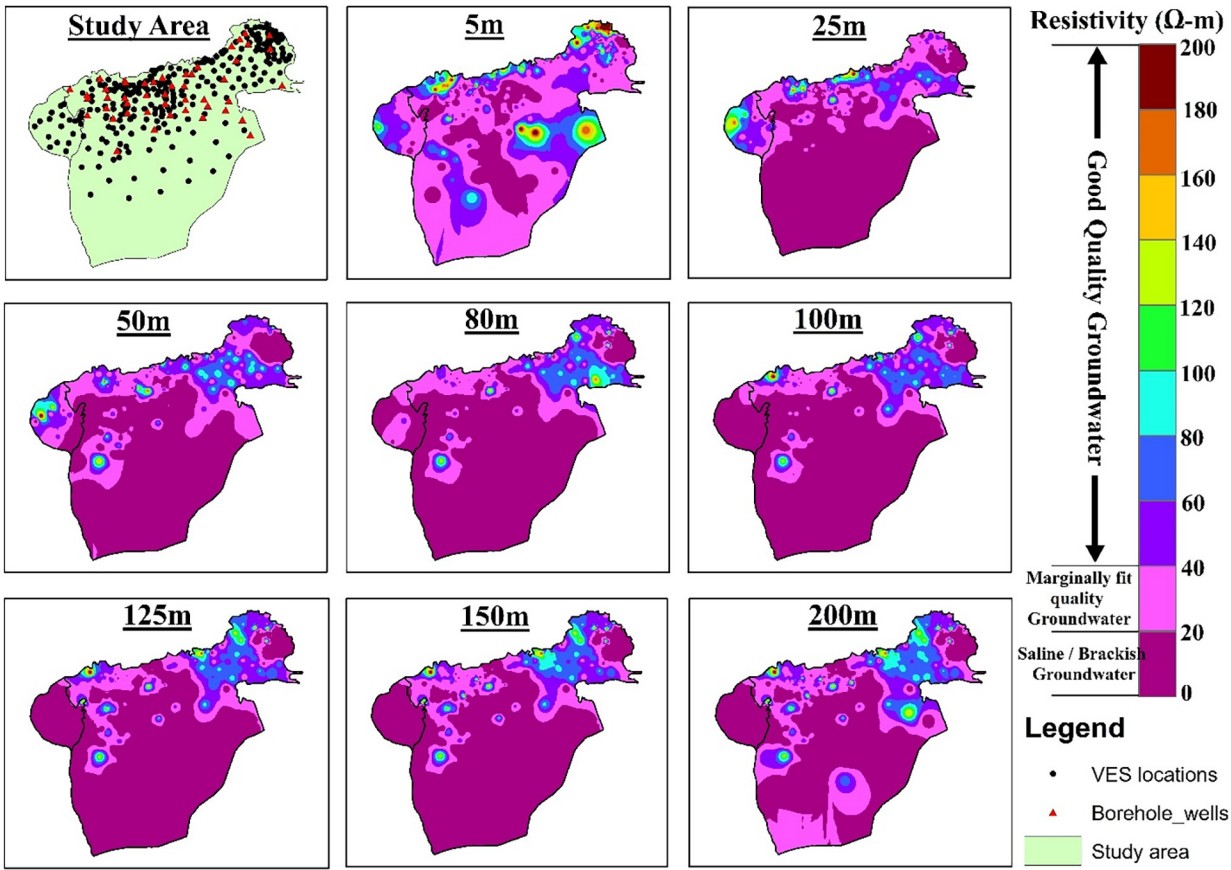

**Fig 10. Visualizing water quality distribution with enhanced precision: Multi-layered maps stacked by resistivity ranges.**

hand, medium to high resistivity values of sand-gravel dominance deposits in the northern segment of the lesser Cholistan strongly indicated the presence of fresh groundwater. The drilling works of water wells for drinking and limited irrigation purposes is recommended within the zones of medium to high resistivities of sand-rich strata. Conversely, the water wells installations within zones of low and very low resistivities are interpreted to yield saline and marginally fit / brackish quality groundwater respectively. Thus, groundwater exploitation within these zones is not recommended on the basis of present research work.

## Supporting information

**S1 Data.**
(XLSX)

**S2 Data.**
(XLSX)

## Author Contributions

**Investigation:** Kashif Arif.

**Methodology:** Kashif Arif.

**Software:** Kashif Arif.

**Supervision:** Perveiz Khalid.

**Writing – original draft:** Kashif Arif.

**Writing – review & editing:** Perveiz Khalid.

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
