## [Decision Letter · Decision Letter 0]

3 May 2023

PONE-D-23-07514Hydrostratigraphy and Hydrogeophysical studies to delineate fresh and saline aquifer boundaries in Lesser Cholistan of PakistanPLOS ONE

Dear Dr. KHALID,

Thank you for submitting your manuscript to PLOS ONE. After careful consideration, we feel that it has merit but does not fully meet PLOS ONE’s publication criteria as it currently stands. Therefore, we invite you to submit a revised version of the manuscript that addresses the points raised during the review process.

We look forward to receiving your revised manuscript.

Kind regards,

Vasanthavigar Murugesan, M.Sc., Ph.D.,https://orcid.org/my-orcid?orcid=0000

Academic Editor

PLOS ONE

Journal Requirements:

Application of electrical resistivity inversion to delineate salt and freshwater interfaces in quaternary sediments of northwest Himalaya, Pakistan - https://doi.org/10.1007/s12517-018-3471-0

In your revision ensure you cite all your sources (including your own works), and quote or rephrase any duplicated text outside the methods section. Further consideration is dependent on these concerns being addressed.

4. Please note that PLOS ONE has specific guidelines on code sharing for submissions in which author-generated code underpins the findings in the manuscript. In these cases, all author-generated code must be made available without restrictions upon publication of the work. Please review our guidelines at https://journals.plos.org/plosone/s/materials-and-software-sharing#loc-sharing-code and ensure that your code is shared in a way that follows best practice and facilitates reproducibility and reuse.

Additional Editor Comments (if provided):

This manuscript's quality could be improved. As a result, your manuscript needs extensive rewriting.

Reviewers' comments:

Reviewer's Responses to Questions

**Comments to the Author**

1. Is the manuscript technically sound, and do the data support the conclusions?

Reviewer #1: No

Reviewer #2: Yes

Reviewer #3: Partly

Reviewer #4: Yes

Reviewer #5: Partly

Reviewer #6: Partly

Reviewer #7: Yes

2. Has the statistical analysis been performed appropriately and rigorously? 

Reviewer #1: N/A

Reviewer #2: N/A

Reviewer #3: No

Reviewer #4: N/A

Reviewer #5: Yes

Reviewer #6: Yes

Reviewer #7: N/A

3. Have the authors made all data underlying the findings in their manuscript fully available?

Reviewer #1: No

Reviewer #2: Yes

Reviewer #3: No

Reviewer #4: Yes

Reviewer #5: Yes

Reviewer #6: Yes

Reviewer #7: No

4. Is the manuscript presented in an intelligible fashion and written in standard English?

Reviewer #1: No

Reviewer #2: Yes

Reviewer #3: Yes

Reviewer #4: Yes

Reviewer #5: Yes

Reviewer #6: Yes

Reviewer #7: Yes

5. Review Comments to the Author

Reviewer #1: Line 34: sources of freshwater (Hasan, 2017), It should be Hasan et al., 2017

Line#41: main source of groundwater contamination (Rasool, 2017), It should be Rasool et al., 2017

Line #47: dunes of low height (Farooq et al., 47 2001). Not included in the reference list

Line#58: Geyh and Ploethner, 2008. Please check the year. Not matched with the given reference list

Line#66: Anwar and Bureste, 2011. Not included in the reference list

Line# 68,69: Cited so many references without mentioning any particular findings.

Line #93 : Kemal et al., 1965. Not included in the reference list

Please recheck the References, all reference list must be cited in the text and vice versal and should be written in the correct format

Line 161-162

It is mentioned that ‘’ VES data is interpreted by using a combination of curve matching technique and computer iterative modeling to find out number of geo-electric layers in the study area” but nothing is mentioned about the curve matching technique.

Line 168-172

For calibration and verification of VES results with borehole data, some of the VES points are acquired near to the existing boreholes/tubewells. Please explain briefly how you correlate lithology with true resistivity values.

All the VES data points were modeled (Fig.5) by using computer software IP2Win (IPI2WIN-1D computer program. 2000; Zananiri et al., 2006; Sultan et al., 2009; Akhter et al., 2012; Farid et al., 2013; Farid et al., 2014; Muhammad and Khalid, 2017). The cited references seem to be irrelevant , it is better to remove these references.

Line 175-177

In general calibration between lithology and 176 resistivities is established using available borehole information and resistivity data and is shown in 177 Table 1

Result and discussion Line 244-255:

Resistivity points (AH-38, YZ- 245 52, and YZ-58, KH-22, AH-40, and HS-22 – 25, YZ-68,69,70,71, and YZ-73) are discussed with reference to water quality but its location is not shown in Fig.1. It is suggested that VES, borehole and test hole should be labeled in Fig.1

There is no agreement of interpreted lithologies between Figs. 6,7,9 and 10 in the light of calibration given in Table 1. In addition, so many figures are presented in the manuscript but in conclusion, nothing is written/mentioned. Conclusions mostly consisting of theory and no quantitative results in conclusions.

In Fig.5, the depth of investigation (AB/2) of YZ-07 is 100 m and consists of 5 layers whereas in Fig.6 the interpreted is 200 m depth and consists of 4 layers.

Results and discussions are very poor. Scientific arguments are very poor in the discussion. Duplication is very common.

Reviewer #2: This study evaluates the fresh and saline aquifer boundaries in Lesser Cholistan of Pakistan. The primary objective of this work is to delineate fresh and saline water interfaces and to map the freshwater aquifer boundaries. However, there are some questions that need to be clarified and improved.

1. All the resolution of the figures should be improved.

2. Abstract should be revised as the authors should give brief introduction in 1 or 2 lines why they are doing this research? They may add a sentence that the present study is intended to identify hotspots for fresh water zones or ………..

3. Line 121 – 123 An updated reference should be added to strengthen the statement. (Rainwater and the Sutlej River are the main recharge sources …………………………. The recharge rate can be divided into deep and shallow categories)

4. In the methodology section, there are many recent research studies available on groundwater, however the authors have added only one 2022 reference. It is good to add some fresh references in this section.

5. What’s the time scale of the calibration and validation of model?

6. Finally, the author should give us recommendations, based on the results for future studies specifically for the study area. Do the authors recommend skimming wells for the area? If yes what should be the locations? The extraction of freshwater layers is used for which purpose? Agriculture? Drinking? Or any other use?

Reviewer #3: This could be a great contribution to an area with a great need for information on groundwater quality. The figures could use a lot of work. I can't really follow the results and conclusions reached from the maps--for example I don't know where the rivers are or the paleochannels that control the resistivity values. I'm not clear how the author differnetiates lithology from water salinity for some layers--for example how tell a clay from a sand containing saline water as both have low resistivity...How are the resistivity value cutoffs for each lithology/water salinity determined? It seems like you would need at least a couple wells with both lithology logs and rsistivity logs so you could assign a resistivity value to a lithology

Reviewer #4: A well written paper with the results presented and discussed adequately. Followings are the points that need to be reviewed:

Line 60 Fig. 1: Base map of the study area showing VES and Borehole locations. Include the VES point numbers and the borehole numbers that you have mentioned in the paper.

Line 105 Kanker : Define what is kanker here?

Line 179 Table 1: Mentions the VES points and Boreholes used for calibrating the results given in the table.

Line 220 - 223 : Statement requires clarity with proper referring to figure or citation.

Line 263: Fig. 7a-h: Maps depicting the distribution of true resistivity across varying depths in two

264 dimensions. (the values in figures are not readable)

Line 272: (resistivity values between VES sites KH-22 and HS-25) these are not readable on figure.

Line 288: Fig. 8: Base map of the study area showing VES points, boreholes, and cross section AA’ (Label the VES points and borehole numbers also on this map.

Line 290: Fig. 9: a) Electrical resistivity distribution versus depth along cross section AA’, (vales along x axis are not readable).

Line 305: No clear boundary was marked or shown on the map only the fresh and saline water zones have been delineated , rephrase the statement accordingly

Reviewer #5: The manuscript entitled: Hydrostratigraphy and Hydrogeophysical Studies to delineate fresh and saline aquifer boundaries in Lesser Cholistan of Pakistan is an original contribution and has not been published in any journals. The present work is the first geophysical study based on the Electrical Resistivity Meter method in the Lesser Cholistan area. The authors used detailed survey data for the present study. As a reviewer, I strongly recommend including this study in the scholarly world. An electrical Resistivity survey is not an easy job in my experience. I do like to admire the authors for initiating such a hectic job. During the review, I found some issues that are explained under the following points:

1. Abstract is very simple and needs to modify as per standard. Include some results value.

2. Collect irrigated water data from the irrigation department and correlates it with cultivated land through surface water. This data will provide more support to your study due to climate change issues, so it is necessary to educate the administration and people regarding water each drop's significance. Groundwater contamination is directly proportional to low water.

3. In line 62 where you wrote study area is part of the Upper Indus Basin, while another study showed Lesser Cholistan as part of the Lower Indus Basin (https://link.springer.com/article/10.1007/s00704-016-2007-3/figures/1)

4. In the Introduction, use recent citations and write the significance of the geophysical study to the context of the Lesser Cholistan area.

5. Clearly express and justified your objectives of the present study. Everyone knows Cholistan is a vast desert in Southern Punjab so its groundwater TDS would be high. You explain why this study was conducted and unique.

6. Method is appropriate and clearly explained

7. Result interpretation is better

8. Conclusion needs some comparison among inside spatial data concerning depth and Hydrostratigraphy. Use results value under what outcome has been observed through the present study.

9. Review and resolve English Grammatical issues.

Reviewer #6: This is an interesting study. However, it needs extensive improvement based on:

1. All Figs should be redrawn to make them clear

2. please, list all VES models nearby boreholes in Fig 6.

3. All results are completely based on VES models, which do not accurate since true resistivity (shown by blue line in Fig 5) does not match with apparent resistivity data in most of models in Fig 5, such as Fig 5a, b, e and f. Please, redraw and correct.

4. Y axis scale in Fig 5 is not clear. Please, make y-axis clear like x-axis scale

5. Please simplify resistivity range in Table 1, i.e., only use less than for very low and greater than with very hgih

Reviewer #7: I appreciate the effort of the author/s that collected the huge geophysical data but need to elaborate more on the linkage between stratigraphy and hydrogeophysical study. Other comments are as follows:

‘The source of hazardous pollutants’ does not explain the type of pollutant found in the entire study area (line 49).

The statement mentioned in lines 99 and 100 is not matched the information illustrated in Fig 2.

Some text in the Figures is not clear and readable, especially in Fig 9.

Fig 1 and 8 is repetition, the section may also be marked in Fig 1.

The sequences of Figs numbers (7 and 8) mentioned in the text are not in order.

Can merge Fig 7 & 9 (because of almost the same information).

The borehole identification number must be mentioned.

Formatting (Line 369)

References found in the list but not cited in the text

Samson (Line 316)

Beaumont (Line 319)

Bhutta (Line 320)

Bowling (Line 323)

Burbank (Line 328)

Burbank (Line 331)

Burbank (Line 334)

Curell (Line 341)

Geyh 2015 (Line 354)

Kamal (Line 361)

Khalid (Line 363)

Muhammad (Line 374)

Pakistan water partnership (Line 385)

Qureshi (Line 387)

Qureshi (Line 390)

Rasool (Line 393)

Sultan 2014 (Line 411)

Yang (Line 426)

Yu (Line 429)

References cited in the text but not found in the list

Rasool (Line 41)

Farooq (Line 46)

Geyh and Ploethner 2008 (Line 58)

Farooq (Line 63)

Anwar and Bureste (Line 66)

Farid (Line 80)

Kemal (Line 93)

Farooq (Line 134)

Louis (line 145)

IPI2WIN-1D computer program. 2000 (lines 170-171)

Sultan (line 171)

Akhter (line 171)

Muhammad and Khalid (line 172)

6. PLOS authors have the option to publish the peer review history of their article (what does this mean?). If published, this will include your full peer review and any attached files.

Reviewer #1: No

Reviewer #2: No

Reviewer #3: **Yes: **Janice Gillespie

Reviewer #4: No

Reviewer #5: No

Reviewer #6: **Yes: **Dr Muhammad Hasan from Institute of Geology and Geophysics Chinese Academy of Sciences

Reviewer #7: No

---

## [Author Response · Author response to Decision Letter 0]

14 Jul 2023

The authors are very thankful to the editor and the reviewers for their valuable time and comments. We have revised the manuscript in the light of the suggestions and comments of the reviewers and the editor. Please see the Response to Reviewers.docx. All important changes made in the manuscript are highlighted in RED COLOR text in the manuscript file. 

The editor comments regarding Copyright of some of the images are also addressed in cover letter. There is no Figure or image which fall in copy right

---

## [Decision Letter · Decision Letter 1]

11 Sep 2023

Hydrostratigraphy and Hydrogeophysical studies to delineate fresh and saline aquifer boundaries in Lesser Cholistan of Pakistan

PONE-D-23-07514R1

Dear Dr. KHALID,

We’re pleased to inform you that your manuscript has been judged scientifically suitable for publication and will be formally accepted for publication once it meets all outstanding technical requirements.

Kind regards,

Venkatramanan Senapathi, Ph.D.

Academic Editor

PLOS ONE

Additional Editor Comments (optional):

Reviewers' comments:

Reviewer's Responses to Questions

**Comments to the Author**

1. If the authors have adequately addressed your comments raised in a previous round of review and you feel that this manuscript is now acceptable for publication, you may indicate that here to bypass the “Comments to the Author” section, enter your conflict of interest statement in the “Confidential to Editor” section, and submit your "Accept" recommendation.

Reviewer #4: All comments have been addressed

Reviewer #5: All comments have been addressed

2. Is the manuscript technically sound, and do the data support the conclusions?

Reviewer #4: Yes

Reviewer #5: Yes

3. Has the statistical analysis been performed appropriately and rigorously? 

Reviewer #4: N/A

Reviewer #5: Yes

4. Have the authors made all data underlying the findings in their manuscript fully available?

Reviewer #4: Yes

Reviewer #5: Yes

5. Is the manuscript presented in an intelligible fashion and written in standard English?

Reviewer #4: Yes

Reviewer #5: Yes

6. Review Comments to the Author

Reviewer #4: The author has addressed all questions and suggestions. The words highlighted have been explained and responses mentioned and incorporated.

Reviewer #5: The authors accepted and inserted reviewers' suggestions in their manuscripts. It should be considered for publication.

7. PLOS authors have the option to publish the peer review history of their article (what does this mean?). If published, this will include your full peer review and any attached files.

Reviewer #4: No

Reviewer #5: No

---

## [Editor Report · Acceptance letter]

26 Sep 2023

PONE-D-23-07514R1 

Hydrostratigraphy and Hydrogeophysical studies to delineate fresh and saline aquifer boundaries in Lesser Cholistan of Pakistan 

Dear Dr. Khalid:

I'm pleased to inform you that your manuscript has been deemed suitable for publication in PLOS ONE. Congratulations! Your manuscript is now with our production department. 

Kind regards, 

on behalf of

Dr. Venkatramanan Senapathi 

Academic Editor

PLOS ONE